# Does Cognitive–Physical Dual-Task Training Have Better Clinical Outcomes than Cognitive Single-Task Training Does? A Single-Blind, Randomized Controlled Trial

**DOI:** 10.3390/healthcare11111544

**Published:** 2023-05-25

**Authors:** Jong-Hyeon Kim, Jin-Hyuck Park

**Affiliations:** 1Department of Occupational Therapy, The Graduate School, Soonchunhyang University, Asan 31538, Republic of Korea; rlawhdgus615@sch.ac.kr; 2Department of Occupational Therapy, College of Medical Science, Soonchunhyang University, Asan 31538, Republic of Korea

**Keywords:** dual-task training, physical exercise, cognitive training, mild cognitive impairment, executive function, instrumental activities of daily living

## Abstract

Purpose: At present, there is a controversy regarding the effect of dual-task training on improving the cognitive function of people with mild cognitive impairment (MCI). This study was to develop and verify the effects of the cognitive–physical dual-task training program on the executive function of older adults with MCI. Method: Participants were randomly allocated to the experimental group (EG) receiving cognitive–physical dual-task training (n = 21) or the control group (CG) receiving cognitive single-task training (n = 21). Results: After 16 sessions for 8 weeks, the Korean version of the Executive Function Performance Task (EFPT-K), the Frontal Assessment Battery (FAB), and Korean version of the Instrumental Activities of Daily Living (K-IADL) tests were implemented to assess people’s executive function and instrumental activities during daily living. As the result, there were no significant differences in general characteristics between both groups (*p* > 0.05). After 16 sessions, the EG showed greater improvements in the EFPT-K (*p* < 0.05; η^2^ = 0.133), the FAB (*p* < 0.001; η^2^ = 0.305), and the K-IADL (*p* < 0.01; η^2^ = 0.221) compared to those of the CG. Conclusion: These results indicate that cognitive–physical dual-task training is clinically beneficial to improve the executive function and daily instrumental activities of older adults with MCI. Cognitive–physical dual-task training is a promising intervention for older adults with MCI.

## 1. Introduction

Since the level of interest in early intervention for dementia has increased, mild cognitive impairment (MCI), a prodromal stage between normal aging and dementia, has gained a lot of attention [1]. People with MCI show deficits in executive function that mainly depend on the prefrontal cortex (PFC), which is one of hallmarks of MCI [2]. Previous studies consistently reported that individuals with MCI exhibit differences in activity in the PFC during cognitive tasks that require executive function compared to that of healthy subjects [3,4].

Therefore, a variety of cognitive interventions for people with MCI have been attempted to improve executive function, but there is inconsistency among effects of cognitive intervention [3]. This inconsistency could be explained by the fact that executive function might not be facilitated when one is conducting a single cognitive task. Executive function could modulate interference between tasks demanding attention at the same time [4]. Therefore, some studies have attempted to investigate effects of cognitive–physical dual task training on executive function [5,6].

Cognitive–physical dual-task training requires coordination between tasks, which is closely correlated with executive function [4,7]. Neuroimaging studies found that executive function heavily depends on the prefrontal cortex [8,9]. Indeed, related studies have consistently reported that a dual task considerably facilitates activation in the prefrontal cortex [10,11], supporting the neuropsychological relationship between dual tasking and executive function. Therefore, dual-task training could be a promising treatment for executive dysfunction in people with MCI who may experience greater challenges in dual-task situations due to the underlying PFC dysfunction. There is a growing body of literature investigating the effects of dual-task training on executive function in MCI [12].

Previous studies reported that cognitive–physical dual-task training had a significant effect on improving executive function in older adults with or without cognitive impairment [12]. Specifically, through training sessions ranging from 12 to 72, the dual-task group outperformed the inactive or active control groups, which performed simple cognitive tasks derived from dual-task programs in executive function. However, compared with active control groups performing conventional cognitive interventions, such as computerized cognitive training, evidence about cognitive–physical dual-task training is limited [9,10]. Therefore, in order to consider dual-task training as an alternative option for improving executive function, it is necessary to compare the effect of dual-task training with cognitive training focusing on executive function.

Moreover, there is a lack of evidence regarding the effects of dual-task training on instrumental activities of daily living (IADLs) improvements [13]. It has been confirmed that IADLs, such as finance management and taking prescribed medicines, comprise various dual-task activities. Additionally, IADLs place high demands on executive function [14], and therefore, IADLs are more likely to deteriorate in people with MCI [15]. Indeed, the previous study reported that impairments in IADL need to be included in the diagnosis of MCI [16]. Accordingly, evidence of dual-task training’s ecological validity by assessing subject’s IADL ability is needed, given that one of the main purposes of cognitive intervention is to provide benefits to subjects.

Therefore, the purpose of the current study was to investigate the superiority of cognitive–physical dual-task training over cognitive training by focusing on executive function in people with MCI. The authors hypothesized that cognitive–physical dual-task training could be more beneficial to improve executive function and IADLs than executive function training is.

## 2. Materials and Methods

### 2.1. Study Design

This study was a single-blind study, and all subjects were randomly divided into the experimental group (EG), who received cognitive–physical dual-task training, or the control group (CG), who performed computerized cognitive training focusing on executive function. Randomization was performed by a blinded experimenter using computer-generated random numbers. Outcome measures were conducted pre- and post-intervention by an assessor who was blinded to group allocation. The intervention consisted of 16 sessions conducted two times a week for eight weeks. This study was approved by the local research ethics committee and registered under the Thai Clinical Trials Registry ID: TCTR20200106005.

### 2.2. Subjects

Older adults with MCI over 65 years were recruited from local community welfare centers in Seoul, South Korea. Subjects were recruited via an author visiting and posting recruitment announcements. They were participating in various programs such as calligraphy and art activities in centers. Recruitment notices were announced for one week, and then 46 subjects who expressed their intention to participate were recruited. The inclusion criteria were in accordance with the previous study [14] and were as follows: (a) they had a subjective memory complaint, (b) they had a Korean version of Montreal Cognitive Assessment (MoCA-K) score lower than 23 [17], and (c) they had an ability to independently perform basic activities of daily living (BADLs). The exclusion criteria were as follows: (a) they had dementia that had been diagnosed by a neurologist, (b) they had neurological, psychiatric, or medical disorders, (c) they had auditory or visual impairments, (d) they participated in cognitive training within the last three months, and (e) they had been educated for less than 6 years. Each subject provided informed consent prior to recruitment into this study, and then voluntarily participated in this study, with the option to stop participating at any time.

The sample size was calculated using G*Power 3.1.7 software (University of Dusseldorf, Dusseldorf, Germany). In accordance with the previous study [18], the effect size was set at 0.60, with power levels at 0.90 and alpha levels at 0.05, resulting in a minimum of 14 subjects being required in each group. The calculated number of subjects was considered to be appropriate considering that previous studies were conducted with 38 subjects [18,19].

### 2.3. Intervention

EG and the CG groups performed cognitive–physical dual-task training and cognitive single-task training, respectively. Both dual-task and single-task training were conducted in community welfare centers and consisted of a total of 16 sessions lasting 45 min a session, two times a week for 8 weeks. Five minutes of rest time was given to both groups in order to minimize their fatigue after 15 min of training. All sessions were conducted by one occupational therapist who has more than 6 years of clinical experience. Subjects individually participated in training sessions and no group session was implemented. Subjects in each group only participated in either single- or dual-task training. To minimize interference, all training sessions were performed in a quiet and designated room.

In this study, dual-task training was designed to improve the executive function of older adults with MCI in accordance with a previous study [19]. The dual-task training program was classified into two categories: a cognitive task and a physical task. This program consisted of simple dual-tasks and more complex dual-task tasks (Table 1). During all sessions, subjects were instructed to simultaneously perform cognitive and physical tasks. The EG performed a total of 16 sessions by repeating eight programs twice in ascending order. In each session, subjects were encouraged to allocate their attention between two tasks as efficiently as possible to maximize their performance on each task, which requires the ability to suppress interference between tasks.

Single-task training was conducted using RehaCom software (Hasomed, Germany), a computerized cognitive program. Subjects sat on a comfortable chair in front of a desktop monitor and were asked to use the input device of the RehaCom to conduct single-task training. The RehaCom contains several cognitive domains such as attention, memory, and executive function. Out of these domains, training contents in the executive function domain were used. The executive function domain has several training contents with different levels of difficulty, which can be automatically adjusted based on the subjects’ performance. During all sessions, the therapist sat next to each subject, observed subject’s performance, and provided instant feedback to each subject.

### 2.4. Measurement

Outcome measures were carried out before and after the intervention. The blinded assessor, who is an occupational therapist with 4 years of experience in using the outcome measures, conducted all outcome measures in a fixed order.

Executive function was assessed by using the Korean version of Executive Function Performance Test (EFPT-K) and the Frontal Assessment Battery (FAB) tests. The EFPT-K was developed to evaluate executive function by directly observing the levels of assistance needed to complete four tasks, such as cooking, making a phone call, taking a medicine, and paying bills. The EFPT-K evaluates executive functions and consists of initiation, organization, sequencing, safety, judgment, and completion during performing the four tasks. The EFPT-K uses five standardized levels of the cueing system from 0 (no cue required) to 5 (dependent). Its scores range from 0 to 100, and a higher score indicates more severe executive dysfunction. Its inter-rater reliability and internal consistency were 0.91 and 0.94, respectively [20]. The FAB is a minor tool that could be used at a bedside or in a clinical setting and consists of six items, such as intellectualization and abstract thinking, thinking flexibility and verbal fluency, motion planning, and reaction to extraneous intervention, inhibitory regulation, and automaticity. Each item is scored from 0 to 3, and the score ranges from 1 to 8, with higher scores indicating better frontal lobe function. The FAB has good concurrent validity (0.82) and inter-rater reliability (0.87) [21].

To assess IADL, the Korean Instrumental Activities of Daily Living (K-IADL) test, which consists of 11 items (shopping, transportation, the ability to handle finances, housekeeping, preparing food, the ability to use a telephone, responsibility for one’s own medication, recent memory, hobbies, watching television, and fixing things around the house) was used. Each item has a Likert scale ranging from 0 to 3 points. Its score ranges from 0 to 33 points, and the higher the scores that subjects obtain art, the more independent in IADL they are. Its internal consistency (Cronhach’s alpha = 0.96) and the test–retest reliability (r = 0.94) are high [22].

### 2.5. Statistical Analysis

All data were analyzed by using SPSS for Windows (version 22.0) in this study. The normal distribution of general characteristics and outcome measures of the subjects were confirmed using the Shapiro–Wilk test. In order to compare the general characteristics of the subjects between the EG and the CG, Chi-square test and independent *t*-test were used. After the 16 sessions, differences in executive function and IADL between both groups were compared by using repeated measures analysis of variance (ANOVA). The effect size (ES) of each intervention group was calculated using a partial η^2^ value. Partial values between ≥0.01 and <0.06, between ≥0.06 and <0.14, or η^2^ ≥ 0.14 were considered to have small, moderate, and large effect sizes, respectively. All statistical significances were accepted at *p* < 0.05.

## 3. Results

### 3.1. Subject’s Characteristics

A total of forty-three subjects were screened by the blinded assessor; two subjects were excluded as their MoCA-K score was 23 or higher, and two subjects declined to participate, resulting in 42 being selected in total (Figure 1). There were no statistically significant differences in the general characteristics between both groups (Table 2).

### 3.2. Executive Function

After 16 training sessions, the experimental group achieved better performances on the EFPT-K and the FAB. Specifically, repeated measure ANOVA showed that the group × time interaction was significant in both the EFPT-K (*p* < 0.05; η^2^ = 0.133) and the FAB (*p* < 0.001; η^2^ = 0.305). These results indicate that there were greater improvements in executive function resulting from dual-task training compared to those in the single-task training group (Table 3).

### 3.3. Instrumental Activities of Daily Living

After 16 training sessions, the experimental group showed better performances on the K-IADL. Specifically, there was a significant group × time interaction in the K-IADL (*p* < 0.01; η^2^ = 0.221). This finding suggests that the dual-task training group achieved more clinical improvement in IADLs compared to that of the single-task training group (Table 3).

## 4. Discussion

The purpose of the current study was to identify the effects of the cognitive–physical dual-task training on executive function and IADLs in older adults with MCI. The dual-task training group showed more significant improvements in both executive function and IADLs over 16 sessions compared to those of the single-task training group. These findings might suggest that cognitive–physical dual-task training could be considered as an effective intervention for executive function and IADLs in older adults with MCI, which is consistent with the results of the meta-analysis study [13].

A variety of interventions have been found to be beneficial and to improve cognitive function in people with or without a cognitive impairment. Among the various interventions, previous studies suggested that physical exercise has a significant effect on cognitive function [23,24]. Indeed, physical exercise has been often implemented in interventions that target elderly people for improving cognitive function [25]. Accordingly, dual-task training, a kind of cognitive intervention combined with physical exercise, has attracted a lot of attention for improving the cognitive function of older populations [12,26].

Under dual-task training, usually, two or more tasks are simultaneously conducted, which requires subjects to modulate competition between cognitive and physical tasks using executive function, such as divided attention and selective attention. Indeed, dual-task training was identified to be correlated with executive function. This is supported by previous studies reporting that people with poor executive function showed lower levels of performance on dual-tasks [27,28]. In addition, a neuroimaging study showed that dual-task training could cause activation in the prefrontal cortex, a brain area that mainly exerts executive function, thereby supporting the correlation between dual-tasks and executive function [29]. In a previous study, subjects showed an improvement in executive function with lower levels of activity in the prefrontal cortex after 16 sessions of dual-task training. This finding suggested that dual-task training could induce an increase in the neural efficiency of the prefrontal cortex [10]. Thus, executive function could be enhanced through the repeated practice of dual-tasks, which is consistent with the findings of this study [14,18].

Meanwhile, in this study, single-task training was selectively conducted using executive function items of the computerized program, which differs from the methodology of previous studies. Most of the previous studies on dual-task training compared the effects of dual-task training with those of non-active training such as single-task training [12,13]. Since single-task training, such as cognitive education programs and treatments, usually does not target executive function, evidence regarding the effects of dual-task training on executive function has been limited [13]. Therefore, in order to affirm the superiority of the effect of dual-task training on executive function, cognitive training focusing on executive function as an active training component needs to be compared. In this study, the superiority of dual-task training was demonstrated because it was proven that the effects of dual-task training were superior, even though it was compared with cognitive training focusing on executive function.

On the other hand, dual-task training showed a greater improvement in people’s instrumental activities during daily living compared to that of single-task training. Instrumental activities of daily living impairments in older adults with MCI could be directly linked with a loss of independence, inducing a caregiver burden since many daily activities are performed by caregivers [30]. Therefore, one of the goals of helping elderly people with MCI is to improve the independence of instrumental activities during daily living, which is consistent with these results. Instrumental activities during daily living, such as shopping, using a telephone, and managing medication, require more complex cognitive skills, such as planning, monitoring, and executing goal-directed activities, which more heavily depend on executive function than basic activities of daily living do [11]. Indeed, previous studies indicated significant relationships between instrumental activities during daily living and executive function [30,31]. Therefore, enhancing executive function might significantly contribute to performing instrumental activities during daily living [32], which is consistent with the current study. In particular, contrary to previous studies investigating effects of dual-task training via using neuropsychological assessment for executive function, this study confirmed that the effects of dual-task training were generalizable to everyday activities, which has clinical significance in terms of ecological validity [33].

This study has some clinical implications. Firstly, this study is clinically relevant as it investigated the effects of cognitive–physical dual-task training, which is commonly used in clinical practice. The findings of this study could provide valuable information for clinicians and patients to make informed decisions about treatment options. Secondly, this study adds to the growing body of literature on dual-task training and could contribute to the development of future guidelines and recommendations. Finally, while statistical significance is an important measure of the strength of the relationship between dual-task training and executive function, this study also addressed the clinical significance of the findings. This study reported the effect size and confidence intervals to provide a better understanding of the magnitude of the training’s effect, and this study used the IADL measure, which is relevant to patients. In addition, compared to the author’s other papers, this study focused more on investigating whether dual-task training helps improve executive function and whether this benefit could transfer to daily life, which makes this study unique.

This study, however, has several limitations. Firstly, the long-term effects of dual-task training were not investigated. Secondly, there was no evidence of changes in activation in the prefrontal cortex using neuroimaging devices. Thirdly, all subjects were motivated enough to voluntarily attend welfare centers, which positively affect the findings of this study. Therefore, these findings need to be interpreted with caution. Fourthly, this study did not directly investigate the effects of examination length on the outcomes, as this was not one of the variables. However, this study took several steps to minimize the potential impact of examination length on the findings of this study. For example, all subjects underwent the same examination protocol. Additionally, this study used measures to ensure that subjects were comfortable and well rested during the examination, which can help to minimize the potential impact of fatigue or discomfort on the results. Finally, the sample size in this study was relatively small. Therefore, neuroimaging studies with a larger sample size and follow-up assessments need to be conducted in order to clarify the robustness of the findings of this study.

## 5. Conclusions

It was confirmed that dual-task training demands more executive function challenges. These findings suggest that dual-task training for older adults with MCI who have difficulties in everyday activities due to executive dysfunction might be more effective than cognitive single-task training is. It is desirable that the easy-to-use, low-cost, and effective training is widely used for older adults with MCI in clinical settings. Future follow-up studies are required to confirm whether the effects are associated with delayed onset of AD in older adults with MCI.

## Figures and Tables

**Figure 1 healthcare-11-01544-f001:**
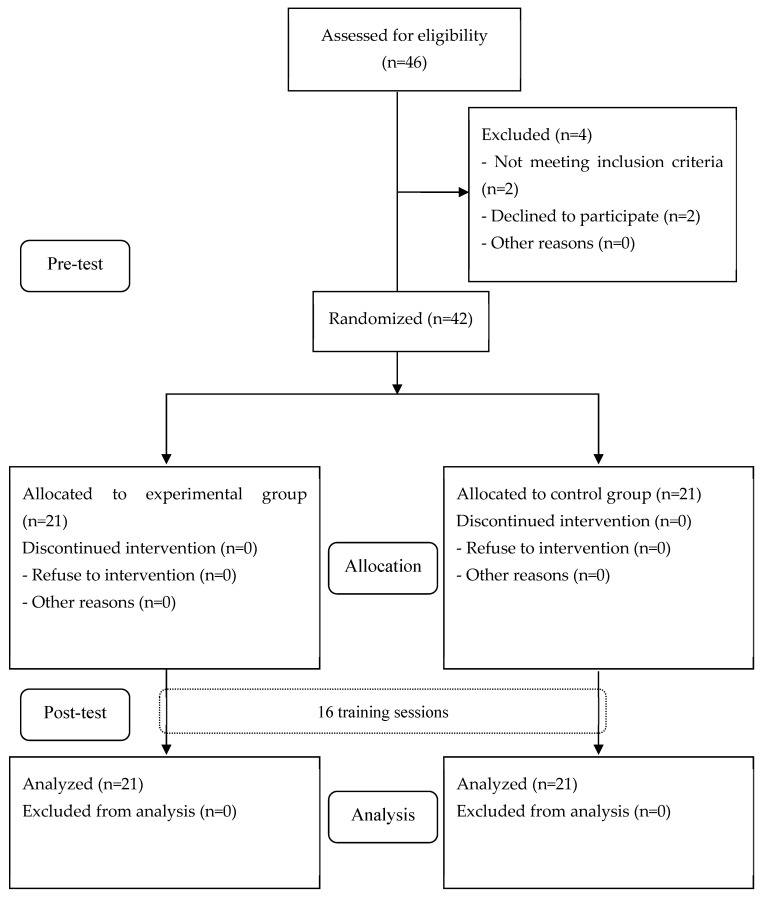
Flow diagram of subjects in the study.

**Table 1 healthcare-11-01544-t001:** Cognitive–physical dual task program.

Session	Theme	Contents
1, 2	Cognitive task	Counting numbers (1 → 100, sequentially) Naming pictures (flowers and fruit)
	Physical task	Aerobic exercise (ROM exercise of wrist, elbow, and shoulder) Strength exercise (pull a thera-band using a hand)
3, 4	Cognitive task	Counting numbers (1 → 100, random) Naming pictures (animals and vegetables)
	Physical task	Aerobic exercise (ROM exercise of wrist, elbow, and shoulder) Strength exercise (pass the ball to the side, back, and forward)
5, 6	Cognitive task	Calculation (addition) Naming them backwards and find the common pictures (flowers and fruits)
	Physical task	Aerobic exercise (ROM exercise of wrist, elbow, shoulder, ankle, knee, and hip) Strength exercise (throw a ball)
7, 8	Cognitive task	Calculation (subtraction) Naming them backwards and find the common pictures (animals and vegetables)
	Physical task	Aerobic exercise (ROM exercise of wrist, elbow, shoulder, ankle, knee, and hip) Strength exercise (pull and push thera-band using a leg)

**Table 2 healthcare-11-01544-t002:** General characteristics of both groups (n = 42).

Characteristics		Experimental Group (n = 21)	Control Group (n = 21)	χ^2^/t
Sex	Male	10 (52.4%)	9 (42.9%)	0.757
	Female	11 (47.6%)	12 (57.1%)	
Age, years (SD)		74.33 ± 5.39	74.71 ± 5.55	−0.255
Education period (years)		8.00 ± 2.38	8.05 ± 3.02	−0.057
MoCA-K (scores)		20.38 ± 1.28	19.57 ± 1.69	1.748

Values are expressed as mean ± standard deviation. MoCA-K, Korean version of Montreal Cognitive Assessment

**Table 3 healthcare-11-01544-t003:** Comparison of executive function and instrumental activities of daily living (n = 42).

	Experimental Group (n = 21)	Control Group (n = 21)	Between-Group Differences	*p*	η^2^
EFPT-K (scores)					
Pre-intervention	53.90 ± 4.17	53.86 ± 4.16	2.81 (0.53; 5.08)	<0.001	0.132 **
Post-intervention	46.62 ± 5.05	49.38 ± 4.28			
Within-group changes	7.28 ± 4.45 (5.25; 9.31)	4.47 ± 2.62 (3.28; 5.66)			
FAB (scores)					
Pre-intervention	9.90 ± 1.13	10.10 ± 1.37	0.85 (0.44; 1.27)	<0.001	0.305 ***
Post-intervention	11.00 ± 1.58	10.33 ± 1.24			
Within-group changes	−1.09 ± 0.70 (−1.41; −0.77)	−0.23 ± 0.62 (−0.53; 0.04)			
K-IADL (scores)					
Pre-intervention	16.52 ± 2.80	16.86 ± 2.33	2.42 (0.97; 3.88)	<0.001	0.221 **
Post-intervention	20.14 ± 2.43	18.05 ± 2.31			
Within-group changes	−0.3.61 ± 3.18 (−5.06; −2.19)	−1.19 ± 0.87 (−1.58; −0.79)			

Data are shown as mean ± standard errors and mean (95% confidence interval) for within and between-group changes. (** *p* < 0.01; *** *p* < 0.001).

## Data Availability

The data presented in this study are available on request from the corresponding author. The data are not publicly available because they are part of an ongoing project.

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
