# Peer review of "Does Cognitive–Physical Dual-Task Training Have Better Clinical Outcomes than Cognitive Single-Task Training Does? A Single-Blind, Randomized Controlled Trial"

_healthcare, 2023, doi:10.3390/healthcare11111544_

Round 1
Reviewer 1 Report
REVIEW:
First of all, I would like to congratulate the authors of the study. I believe that the study of mild cognitive impairment is very important for the scientific community. The increase in MCI should lead to studies of this calibre, as its incidence increases exponentially as people's life expectancy increases. I would like to congratulate the authors of the study for their scientific rigour, as well as for the exposition of everything related to the study.
I believe it should be considered for publication in the current format. But not before making some minor changes.
I add the following comments:
Novelty: The study is sufficiently original to be considered for publication in the journal.
Scope: The topic fits the scope of the journal, taking into account that it is an analysis of a possible area of action in the health sciences.
Relevance: The data are adequately interpreted, and are sufficiently scientifically sound to be relevant. The sample size is optimal, but perhaps it should be explained a little more how the sample size shown was achieved. In case it was a convenience sample it should be stated, and similar studies should be found to provide data for this possible sample size. I would ask for a more rigorous description of the method as well as the procedure used. Special emphasis should be placed on a more detailed description of the variables under study, as well as the description of the statistical analysis performed. The "methods" section could be improved.
Quality: The article is adequately written, well structured and supported by similar studies that raise issues to be worked on in this project. The "results" section needs to be improved, as I believe that the most important results should be written, not only in tables.
Scientific soundness: The study presents the necessary scientific soundness to extract highly significant and valid results for the scientific community. The sample size is clearly justified and in this justification we think that it is an adequate sample size for the methodological characteristics of the study. I would ask for a more rigorous description of the method as well as the procedure used. Special emphasis should be placed on a more detailed description of the variables under study, as well as the description of the statistical analysis performed. The "methods" section could be improved.
Readership interest: We think that the scientific robustness is sufficient to attract a significant number of readers from the scientific community.
Overall merit: Overall, we think the study could have sufficient scientific strength to be considered for publication. We encourage researchers to pursue the topic further and to increase knowledge on this important topic in the health sciences in future studies.
Level of English: I do not feel qualified to judge on English language and style.
Specific comments:
- Increase sample size description.
- Improve the method section.
- Describe the most important results, not just show them in tables.
Author Response
Thank you for your supportive comment. Our manuscript could be improved after this revision. Please see the attached file for detailed responses to your comment.

Reviewer 2 Report
As author of other 4 papers in the same topic it is recommendable to state if this trial is the same as the ones published elsewhere.
Please described the roles of every member of your team that has allowed you to conduct the trial.
The randomization process should be described.
What's the clinical relevance of the results. Statistic significance is not enough to support the efficacy of an intervention.
Author Response

(The authors gave the same response as above.)

Reviewer 3 Report
1. Would the length of examinatiopn have a role in the results?
2. Dual task issues are a major factor in subcortical involvement such as parkinsonism, and it would be necessary to to know what underlying diagnoses was covered by what the authors call "MCI", which in itself is not a "disease".
3. What would be the proportion of frontal type vs. posterior/parietal predominance dysfunction in the selected subjects for each group? And how would that affect findings?
Author Response

(The authors gave the same response as above.)

Round 2
Reviewer 2 Report
I thank you for your efforts. With the current amendments, I accept the document as it stands.
Author Response
Thank you for your supportive comments. We consider this comment as no further revision is required.